# CogVideo: Large-scale Pretraining for Text-to-Video Generation via Transformers

## Abstract

Large-scale pretrained transformers have created milestones in text (GPT-3) and text-to-image (DALL-E and CogView) generation. Its application to video generation is still facing many challenges: The potential huge computation makes it unafforable for a full training; The scarcity and weak relevance of text-video datasets hinder the model to understand complex movement semantics. In this work, we present 9B-parameter transformer CogVideo, trained by inheriting a pretrained text-to-image model, CogView2. We also propose multi-frame-rate hierarchical training strategy to better align text and video clips. As (probably) the first open-source large-scale pretrained text-to-video model, CogVideo outperforms all publicly available models at a large margin in both machine and human evaluations.

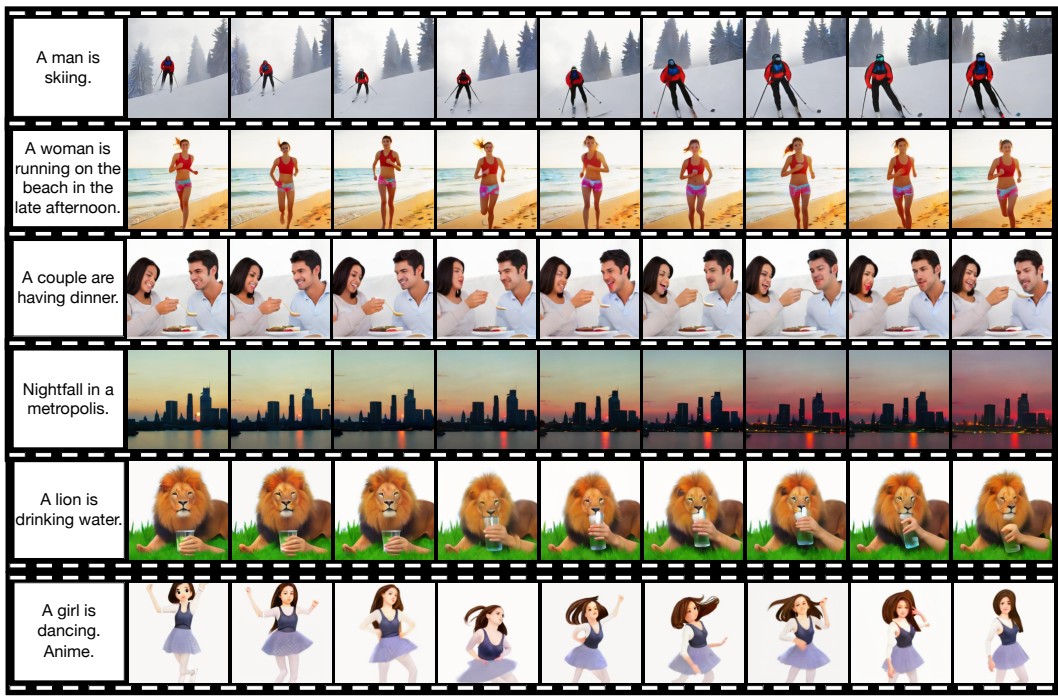

Figure 1: Samples generated by CogVideo. The actual text inputs are in Chinese. Each sample is a 4-second clip of 32 frames, and here we sample 9 frames uniformly for display purpose.

# 1  Introduction

Autoregressive transformers, e.g. DALL-E [19] and CogView [5], have revolutionized text-to-image generation recently. It is natural to investigate the potential of autoregressive transformers on text-to-video generation. Previous works followed this basic framework [36, 9], e.g. VideoGPT [37], verifying its superiority over GAN-based methods [4, 27], but are still far from satisfaction.

One common challenge is that the generated video frames tend to gradually deviate from the text prompt, making the generated characters hard to perform the desired actions. Vanilla autoregressive models might be good at synthesizing videos with regular (e.g. straightly moving cars) or random patterns (e.g. speaking by randomly moving lips), but fail on text prompt such as "a lion is drinking water". The main difference between the two cases is that, in the former case the first frame already provides sufficient information for the subsequent changes, while in the latter the model has to precisely understand the action "drink" in order to correctly generate the desired action — the lion lifts the glass to its lip, drinks and then puts down the glass.

Why do the autoregressive transformers well understand the text-image relations, but struggle to understand the text-action relations in videos? We hypothesize that the datasets and the way to utilize them are the main reasons.

First, it is possible to collect billions of high-quality text-image pairs from Internet [19], but the text-video data are more scarce. The largest annotated text-video dataset, VATEX [32], has only 41,250 videos. The retrieval-based text-video pairs, e.g. Howto100M [17], are weakly relevant and most of them only describe the scene without the temporal information.

Second, the duration of videos varies a lot. Previous models split the video into many clips with a fixed number of frames for training, which destroys the alignment between the text and its temporal counterparts in the video. If a "drinking" video is split into four individual clips of "holding a glass", "lifting", "drinking" and "putting down" with the same text "drinking", the model will be confused to learn the accurate meaning of drinking.

**Present Work.** Here we present a large-scale pretrained text-to-video generative model, CogVideo, which is of 9.4 billion parameters and trained on 5.4 million text-video pairs. We build CogVideo based on a pretrained text-to-image model, CogView2 [6], in order to inherit the knowledge learned from the text-image pretraining. To ensure the alignment between text and its temporal counterparts in the video, we propose the *multi-frame-rate hierarchical training*. The flexibility of the textual condition makes it possible to simply prepend a piece of text describing the frame rate to the original text prompt for modeling different frame rates. To keep the text-video alignment, we choose a proper frame rate description to include the complete action in each training sample. The frame rate token also controls the intensity of the changes throughout continuous frames in generation. Specifically, we train a sequential generation model and a frame interpolation model. The former model generates key frames according to the text, and the latter recursively fill the middle frames by varying the frame rates to make the video coherent. As shown in Figure 1, CogVideo can generate high-resolution (480×480) videos. Human evaluation demonstrates that CogVideo outperforms all publicly available models at a large margin. Our main contributions can be concluded as follows:

- We present CogVideo, which is the **largest** and **the first open-source** pretrained transformer for text-to-video generation in the general domain.

- CogVideo elegantly and efficiently finetunes a text-to-video generative model from a pretrained text-to-image generative model, avoiding the expensive full pretraining from scratch.

- We propose the multi-frame-rate hierarchical training to better align text-clip pairs, which significantly improves the generation accuracy, in particular for movements of complex semantics. This training strategy endows CogVideo with the capacity of controlling the intensity of changes during the generation.

# 2  Related Work

## 2.1  Video Generation

Video generation is a long-standing research topic. Most previous works focus on the next-frame prediction task — forecasting the future frames based on the first video frame. Early works, e.g.

CDNA [8] and PredRNN [33], leverage deterministic methods to directly predict the next frame via CNNs or RNNs. However, these deterministic models are unable to capture the stochastic temporal patterns and synthesize coherent complex scenes. Generative models, especially Generative Adversarial Networks [10] (GANs), begin to dominate the area as they can perform unconditional or class-conditional video synthesis without the first frames. VGAN [31] is the first one to use GAN for video generation. It decomposes video to a static background and a moving foreground, and then generates them with 2D and 3D convolutional networks respectively. TGAN[20] proposes to separately generate the temporal latent variables and spatial information, and MoCoGAN [27] similarly decomposes the latent space into context and motion subspaces. DIGAN [38] applies implicit neural representations for video encoding. Recently, text-to-video generation emerges as a promising direction. The framework of VQVAE [29] and autoregressive transformers [30, 1] quickly becomes the mainstream method [35, 36, 9]. Ho et al. [11] proposes video diffusion model along with a gradient method recently for text-to-video generation. The previous methods are basically trained on a specific dataset, e.g. UCF-101 [23], making the trained model domain-specific. Moreover, most of these models are not publicly available.

## 2.2 Autoregressive Transformer

Recent years have witnessed the autoregressive transformer emerging as a powerful generative model. The autoregressive models become the most prevalent framework for text generation [24]. With its prominent capacity of fitting, transformer [30] gradually becomes the standard neural structure for text generation. One milestone is GPT-3 [1]. In computer vision, van den Oord et al. [29] first proposes to train a VQVAE to compress the image into a sequence of tokens from a learned dictionary, which can be efficiently handled by autoregressive models. VQ-GAN [7] learns a more semantic-aware dictionary for unconditional image generation. In the text-to-image generation, pre-trained autoregressive transformers such as DALL-E [19] and CogView [5] have shown superiority in open-domain image generation. Besides the pure GPT-style generation, CogView2 [6] proposes a new language model CogLM for infilling in the image generation.

Recent autoregressive transformers [18, 37, 35, 36] have also shown their superiority in video generation. Among them, GODIVA [35] and NÜWA [36] focus on the open-domain text-to-video generation. However, they simply generate frames or frame blocks one by one in a chronological order, and may suffer from poor text-video alignment (Cf. § 1).

# 3 Method

In this section, we first introduce *multi-frame-rate hierarchical training* to better align text and video semantics in § 3.1, and then illustrate an efficient method *dual-channel attention* to inherit the knowledge in pretrained text-image models for video generation in § 3.2. To overcome the large memory and time overhead caused by the large model and long sequence, we refer to Swin Attention [14] and extend it to autoregressive video generation in § 3.3.

## 3.1 Multi-frame-rate Hierarchical Training

Here we present the *multi-frame-rate hierarchical training* and generation. We follow the framework of VQVAE [29] and first tokenize each frame into image tokens. Each training sample consists of 5 frame of tokens, but our training method differs in the construction of training sequences and generation process.

**Training.** The key design is to add a frame-rate token to the text and sample frames at this frame-rate to compose a fixed-length training sequence. The motivations are two folds:

(1) Directly separating the long video into clips at a fixed frame-rate often leads to semantic mis-matching. We still use the full text but the truncated clip might only contain incomplete action.

(2) The adjacent frames are usually very similar. A giant change over the previous frame will probably incur a large loss. This will lead the models less inclined to explore the long-range correlation because to simply copy the previous frame acts like a shortcut.

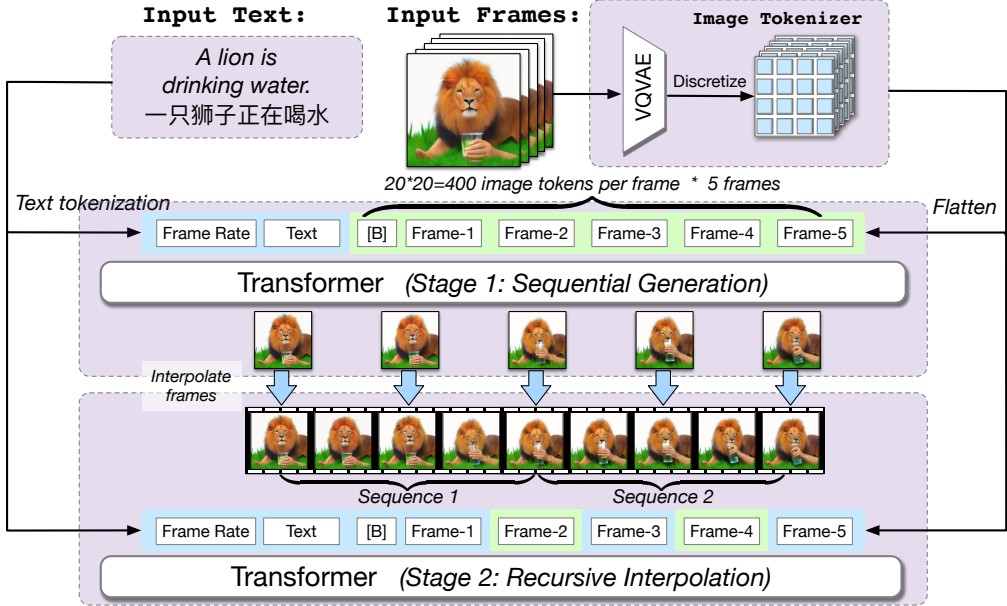

Figure 2: Multi-frame-rate hierarchical generation framework in CogVideo. Input sequence includes frame rate, text, frame tokens. [B] (Begin-of-image) is a separator token, inherited from CogView2. In stage 1, $T_s$ frames are generated sequentially on condition of frame rate and text. Then in stage 2, generated frames are re-input as bidirectional attention regions to recursively interpolate frames. Frame rate can be adjusted during both stages. Bidirectional attention regions are highlighted in blue , and unidirectional regions are highlighted in green .

Therefore, in each training sample we want the text and the frames match as possible. We predefined a series of frame-rates, and select the lowest frame-rate for each text-video pair, as long as we can sample at least 5 frames at this frame-rate in the video.

Although the above method increase the alignment of text and video, the generation at a low frame-rate could be incoherent. We train another *frame interpolation* model to insert transition frames to the generated samples of the sequential generation model. Thanks to the generality of CogLM [6], the two models can share the same structure and training process only with different attention masks.

**Generation** The multi-frame-rate hierarchical generation is a recursive process, illustrated in Figure 2. Specifically, the generation pipeline consists of a sequential generation stage and a recursive interpolation stage:

(1) Sequentially generate $T_s$ key frames based on a low frame rate and text. The input sequence is [{Frame Rate}{Text} [B] {Frame1} ... {Frame $T_s$}]. In practice, we always set $T_s = 5$ and the minimum sampling frame rate to 1 fps.

(2) Recursively interpolate frames based on the text, frame rate and known frames. In each round of interpolation, we split generated frames into multiple $\lceil \frac{T_s}{2} \rceil$-frame blocks overlapping at the beginning and the end, and interpolate a frame between the successive frames in each block. The input sequence is [{Frame Rate}{Text} [B] {Frame1} ... {Frame $T_s$}], where Frame $2i(i = 1, 2, ..., \lfloor \frac{T_s}{2} \rfloor)$ are to be autoregressively generated. By recursively halfing {Frame Rate}, we can conduct finer and finer interpolation to generate videos of many frames.

**The effect of CogLM.** Tasks such as frame interpolation rely heavily on bidirectional information. However, most previous works use GPT [35, 37, 36], which is unidirectional. To be aware of the bidirectional context, we adopt Cross-Modal General Language Model (CogLM) proposed in [6] which unites bidirectional context-aware mask prediction and autoregressive generation by dividing tokens into unidirectional and bidirectional attention regions. While bidirectional regions can attend to all bidirectional regions, unidirectional regions can attend to all bidirectional regions and previous unidirectional regions. As shown in 2, (1) all frames in stage 1 and the 2nd, 4th frames in stage

137 2 are in the unidirectional region; (2) {Frame Rate}, {Text} and all other frames belong to the
138 bidirectional region. In this way, bidirectional attention context is fully exploited in text and given
139 frames without interfering auto-regressive frame prediction.

## 3.2 Dual-channel Attention

141 Large-scale pretraining usually demands a large dataset. For open-
142 domain text-to-video generation, ideally we need the dataset to
143 cover sufficient text-video pairs to infer both spatial and tem-
144 poral correlation between video and text. However, to collect
145 high quality text-video pairs is often difficult, expensive and time-
146 consuming.

147 A natural idea is to make use of the image data to facilitate the
148 learning of spatial semantics. Video Diffusion Model [11] and
149 NÜWA [36] try to add text-image pairs into text-video training,
150 which achieves better results on multiple metrics. However, as
151 for training a video-only generation model, adding image data
152 will significantly increase training cost, especially in large-scale
153 pretraining scenarios.

154 In this paper, we propose to leverage pretrained image generation
155 models instead of image data. Pretrained text-to-image models,
156 e.g. CogView2 [6], already have a good command of the text-
157 image relations. The coverage of the dataset to train these model
158 is also larger than that of videos.

Figure 3: Dual-channel atten-
tion. We initialize Attention-
plus the same as Attention-base
so that the model behaves ex-
actly the same as CogView2
when it is initialized.

159 The proposed technique is *dual-channel attention*, where we only
160 add a new spatial-temporal attention channel to the pretrained CogView2 [6] at each transformer
161 layer. All the parameters in the CogView2 are frozen in the training, and only the parameters in the
162 newly added attention layer(See the Attention-plus in Figure 3) are trainable.

163 Here we also emphasize that directly finetuning CogView2 for text-to-video generation cannot well
164 inherit the knowledge, because the temporal attention follows a different attention pattern and quickly
165 ruins the pretrained weights during the initial phase of training with large gradients.

166 Specifically, a Transformer layer with dual-channel attention can be computed as

$$\hat{x}_l = \text{LayerNorm}(x_l), \tag{1}$$

$$\widetilde{x}_l = \alpha \cdot \text{Attention-base}(\hat{x}_l) + (1 - \alpha), \cdot \text{Attention-plus}(\hat{x}_l), \tag{2}$$

$$x_{l+1} = \text{FFN}(\text{LayerNorm}(x_l + \widetilde{x}_l)), \tag{3}$$

167 where $x_l$ denotes input features of layer $l$; Attention-base and Attention-plus denote two attention
168 channels; FFN and LayerNorm represent Feed-Forward Networks and LayerNorm respectively; $\alpha$
169 is a vector with length of hidden-size and normalized to $(0, 1)$. The whole structure is the same as
170 CogView2 when ignoring Attention-plus.

171 Both channels are computed as normal multi-head attention with a certain receptive field formulated
172 as follows. For token at $(t, x, y)$ in frame block of size $(T_s, X, Y)$ (where $(t, x, y)$ corresponds to
173 coordination along time, height and width dimension), receptive field RF is a 3D block with extent
174 $l_t, l_x, l_y \in \mathbb{N}^+$:

$$\text{RF}_{(t,x,y)} = \{(k, i, j) \mid |x - i| < l_x, |y - j| < l_y, |t - k| < l_t, (k, i, j) \notin \text{Mask}_{(t,x,y)}\}, \tag{4}$$

175 where $\text{Mask}_{(t,x,y)}$ represents CogLM attention mask for token $(t, x, y)$. For Attention-base, we
176 restrict receptive field to current frame, i.e, $l_x = X, l_y = Y, l_t = 1$, to fully use CogView2's spatial
177 modeling ability (therefore referred to as *spatial channel*). For Attention-plus, which is the only
178 new parameters in CogVideo, we set receptive field to a 3D local block throughout the whole time
179 dimension, i.e. $l_x = A_x, l_y = A_y, l_t = T_s$ (therefore referred to as *temporal channel*). $A_x, A_y$
180 are hyper-parameters satisfying $A_x \leq X, A_y \leq Y$. With $A_x$ and $A_y$, CogVideo is able to flexibly
181 trade off between quadratic attention cost and size of receptive field. In practice, we use shifted
182 window attention [15] as a approximation of 3D block attention and extend it to CogLM scenario, as
183 illustrated in subsection 3.3.

It is worth noting that two channels are fused and share the same FFN in each layer, because FFN is a module of heavy parameters containing much vision knowledge. Due to similarity between images and videos, bringing its knowledge to temporal channel will facilitate video modeling. Finally, sharing FFN can reduce parameters, thus speed up training and reduce memory overhead.

### 3.3 Shifted Window Attention in Auto-regressive Generation

To overcome large time and memory overhead in temporal channel during training and inference, we refer to Swin Attention proposed in [14] and extend it to auto-regressive scenario by applying auto-regressive attention mask in shifted windows.

Different from non-autoregressive scenario which original Swin Transformer explores, we propose that Swin Attention can further accelerate auto-regressive inference because of restricted receptive field. As shown in Figure 4, receptive field is restricted by

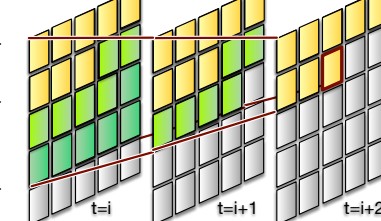

- Auto-regressive mask. A token can only attend to previous frames or tokens before itself in current frame.

- Shifted window. Only tokens within distance of window size in both width and height dimension can be directly attended to.

Figure 4: Receptive field (in yellow or green) for the token in red box. Shifted window size is $2 \times 2$ in this example.

Suppose $X,Y$ is the height and width of each frame, and $A_x,A_y$ are the height and width of shifted window. For two tokens at $(t_1, x_1, y_1)$ and $(t_2, x_2, y_2)$, $t_1 < t_2$, the latter cannot attend to the former either directly or indirectly if

$$(x_1 - x_2)Y + (y_1 - y_2) \geq (t_2 - t_1 + 1)(A_xY + A_y) \tag{5}$$

is satisfied. That is to say, the i-th token in frame $t_1$ can be generated with the $(i - A_xY + A_y)$-th token in frame $t_1 + 1$ in parallel. In this way, we can generate $\lfloor \frac{XY}{A_xY+A_y} \rfloor$ tokens in parallel at most, thus greatly enhance parallelism and accelerate inference compared to auto-regressive with standard attention which can only generate one token at a time.

## 4 Training

Based on methods above, the training details of CogVideo are listed as follows:

**Model.** The backbone of CogVideo in both stages is a Transformer with dual-channel attention. The Transformer has 48 layers, with the hidden size of 3072 in each attention channel, 48 attention heads and 9.4 billion parameters in total. Among them, 6 billion parameters are fixed to CogView2's parameters, which includes Position-wise Feed-Forward Networks (FFN), spatial channel of dual-channel Attention, first frame's positional embeddings and all image and text vocabulary embeddings. The specific implementation of Transformer structure is almost identical to CogView [5] such as using Sandwich LayerNorm and PB-Relax to stablize training. Shifted CogLM attention window is adoppted in recursive interpolation model with window size $10 \times 10$.

**Dataset.** We pretrain our model on a dataset of 5.4 million captioned videos with a spatial resolution of 160x160. For sequential generation model (Stage-1), we adjust frame rate in each sample to accomodate the whole video, while the minimum frame rate is set to 1 fps. For recursive interpolation model(Stage-2), we split videos into clips of different length to accomodate prediction on multiple frame rates including 2,4,8 fps.

**Pretraining.** The sequence lengths in both stages are 2065, consisting of 64 text tokens, 5 (frames) x 400 (per frame) image tokens, and 1 seperator token. Both text and images are tokenized with icetk[1].The parameters are updated by Adam with max learning rate $= 2 \times 10^{-4}$, $\beta_1 = 0.9$, $\beta_2 = 0.95$, weight decay $= 1 \times 10^{-2}$. See Appendix for pretraining details.

---

[1] https://github.com/THUDM/icetk

Table 1: (Left) Video generation performance on UCF-101. Class labels are used as text inputs. * denotes the model is trained on the training split of UCF-101 only. (Right) Video generation performance on Kinetics-600. Metrics are measured on generated videos of 16 frames priming on first 5 frames, following settings in [18]. ** denotes groundtruth used in FVD testing is blurred with our image tokenizer icetk.

| Method | IS ($\uparrow$) | FVD ($\downarrow$) |
|---|---|---|
| VideoGPT[37] | 24.69 | - |
| DVD-GAN[4] | 27.38 | - |
| TGANv2[21]* | 28.87 | 1209 |
| MoCoGAN-HD[25] | 32.36 | 838 |
| DIGAN[38]* | 29.71 | 655 |
| DIGAN[38] | 32.70 | 577 |
| TATS-base[9] | 79.28 | 332 |
| CogVideo (Ours) | 50.46 | 626 |
| CogVideo (Ours)** | - | 545 |

| Method | FVD |
|---|---|
| Latent Video Tranformer[18] | 224.73 |
| Video Transformer[34] | 170 |
| DVD-GAN-FP[4] | 69.15 |
| TriVD-GAN-FP[16] | 25.74 |
| CogVideo (Ours) | 109.23 |
| CogVideo (Ours)** | 59.55 |

## 5 Experiments

### 5.1 Machine Evaluation

Machine evaluation is conducted on two popular benchmarks for video generation, i.e., UCF101 [23] and Kinetics-600 [3]. Following Rakhimov et al. [18], Yu et al. [38], we use Fréchet Video Distance(FVD) [28] and Inception score(IS) [22] as metrics in the evaluation. FVD is calculated based on I3D model[2] trained on Kinetics-400, and IS is based on C3D model [26] which was first trained with Sports-1M dataset [12] and then fine-tuned on the UCF101 dataset. Our evaluation code is the same as the official TGAN-v2 implementation[2].

**UCF-101** is a human action dataset consisted of 13,320 videos annotated with 101 action classes. Due to the image style and frame rate gap between CogVideo's training set and UCF-101, we use class labels as the input text and fine-tune CogVideo on the whole dataset for 10,000 iterations with batch size = 192. During inference, we sample class labels according to the class distribution. FVD and IS are evaluated over 2048 and 10,000 samples respectively, following Yu et al. [38]. Results are shown in Table 1 (Left).

**Kinetics-600** dataset contains 600 classes of human action videos, with roughly 350k train and 50k test videos in total. We use the action category as input text, and fine-tune CogVideo on the training set for 12,000 iterations with batch size of 640. Following the setup of Weissenborn et al. [34], Rakhimov et al. [18], we center-crop and down-sample each frame to 64x64, and measure with FVD. Results are shown in Table 1 (Right).

### 5.2 Human Evaluation

To further evaluate CogVideo, we invite 90 anonymous evaluators to rate for CogVideo and other open-source baselines including GAN-based model TGANv2 [21] and GPT-based model VideoGPT [37]. 30 classes in UCF101 are randomly picked as text conditions, and several aspects are rated (See Appendix for details). For VideoGPT, we use the official unconditonal pretrained model[3] to generate samples. For TGANv2, we use the official source code to train an unconditional generation model under the same setting as that in Saito et al. [21]. To assign unconditionally generated samples into corresponding categories, we choose TSM [13] as the action recognition model and only samples with confidence >80%. Results in Figure 5 show that CogVideo significantly outperforms baselines on multiple important aspects including frame texture, motion realism and semantice relevance, and achieves the top score by overall quality. It can be seen that 49.53% evaluators choose CogVideo as the best method, and only 15.42% and 5.6% favor VideoGPT and TGANv2, respectively.

---

[2]https://github.com/pfnet-research/tgan2
[3]https://github.com/wilson1yan/VideoGPT

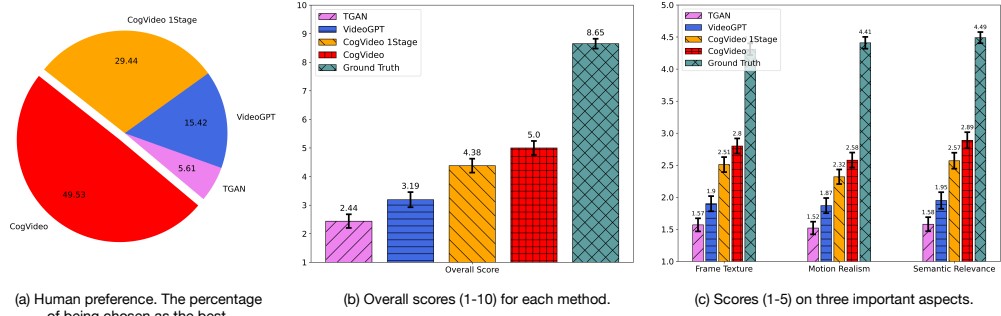

(a) Human preference. The percentage of being chosen as the best.

(b) Overall scores (1-10) for each method.

(c) Scores (1-5) on three important aspects.

Figure 5: Human evaluation results. "CogVideo 1Stage" refers to the method in ablation study, which generates videos sequentially with CogVideo's Stage-1 Model only by recursively reinserting last 2 generated frames into input and generate future frames.

Table 2: Ablation study on a 5,000-sample subset of Kinetcis-600's testset. FVD is evaluated on generated 11-frame samples priming on 5 frames and ground-truth blurred by our image tokenizer. The setting column indicates the difference between each method and CogVideo. Models of each setting are trained on Kinetics-600 trainset for 10,000 iterations with batch size of 320.

| Method | Setting | FVD ($\downarrow$) |
|---|---|---|
| CogVideo | None | 108.27 |
| 1-stage Generation($N_{overlap} = 1$) | $-$ hierarchical | 137.13 |
| 1-stage Generation($N_{overlap} = 2$) | $-$ hierarchical | 120.82 |
| Initialzed to CogView2 | $-$ Pretrain | 124.92 |
| Randomly Initialzed | $-$ Pretrain $-$ CogView | 166.13 |

## 5.3 Ablation Study

To verify the effectiveness of hierarchical multi-frame-rate generation and incorporating CogView2, we conduct ablation study quantitatively and qualitatively on Kinetics-600 and UCF-101 datasets.

**Hierarchical multi-frame-rate generation.** In comparison with CogVideo, we fine-tune a 1-stage video generation model on Kinetics-600 from the sequential generation model in CogVideo, which generates long videos by recursively reinserting last $N_{overlap}$ frames into the input to sample next $N_s - N_{overlap}$ frames. Larger $N_{overlap}$ means more previous frames can be utilized during the inference, but will increase time overhead.

**Dual-channel attention with CogView2's weights.** We additionally train (1) A randomly initialized model; (2) A model incorporating CogView2's weights but leaving temporal channel randomly initialized and unfixed (equivalent to CogVideo without pretraining on videos) on Kinetics-600.

### 5.3.1 Quantitative Evaluation

All aforementioned models have been trained for 11,000 iterations with batch size of 160. Quantitative results are shown in Table 2. We can see that the hierarchical method is clearly superior to 1-stage generation with different $N_s$, and model initialized with CogView2's weights has lower FVD than randomly initialized one.

Figure 6 plots the training loss curve of (1) finetuning CogVideo; (2) training model from random initialization; (3) training model initialized to CogView2 and partially fixed. We can see that CogView2 endows model with a

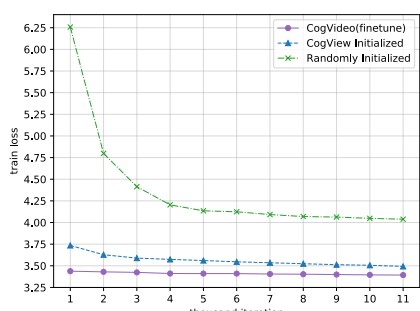

Figure 6: Training loss in ablation study.

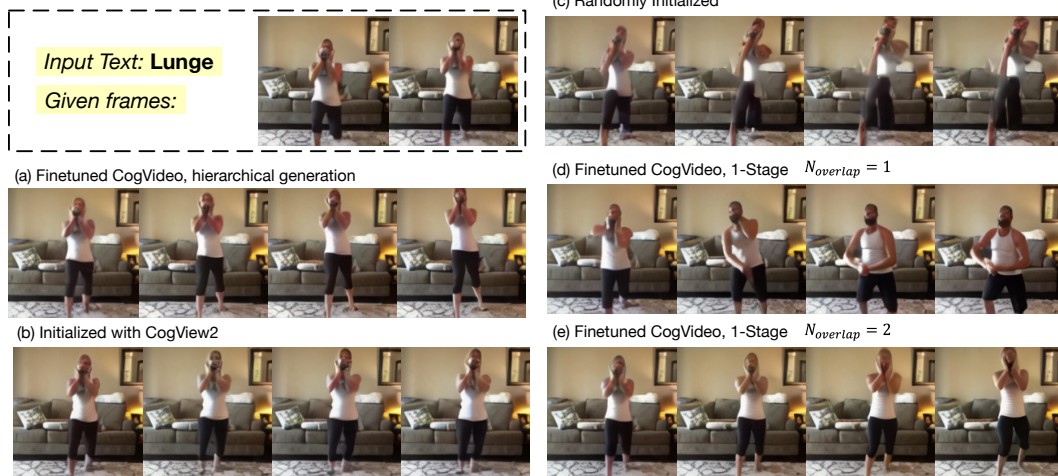

Figure 7: Video samples in ablation study, which are generated priming on class label and first 5 frames in Kinetics-600. All samples are down sampled by extracting one in every three frames for display purpose. (a) Use fine-tuned CogVideo to hierarchically generate samples. (b) Train a model on Kinetics-600 which is initialized as and partially fixed to CogView2, and hierarchically generate samples. (c) Train a model on Kinetics-600 which is randomly initialized, and hierarchically generate samples. (d)(e) Use fine-tuned CogVideo to generate frames in 1 stage with different $N_{overlap}$.

good initialization point from which the loss function can converge faster to a lower value. Also, fixing part of the parameters to CogView2 reduce optimization cost, which gains more than 2x acceleration when using optimization CPU-offload mode in deepspeed.

### 5.3.2 Qualitative Evaluation

Qualitative comparison is shown in Figure 7. While model trained from random initialization tends to produce irrational deformation, model incorporating CogView2 is able to model objects better. And samples generated hierarchically performs better on content consistency and motion rationalization.

We also conduct human evaluation between 1-stage and hierarchical video generation model under the same setting as 5.2. As shown in 5, hierarchical model, i.e. CogVideo, outperforms 1-stage model on semantic relevance, motion realism as well as texture quality. This is probably because 1-stage model tends to constantly generate small movements which make the whole video unrealistic, and if one generated frame collapses, the subsequent frames often suffer from severe degradation.

## 6 Conclusion

We present CogVideo, to the best of our knowledge, the largest and the first open-source pretrained transformer for text-to-video generation for the general domain. CogVideo is also the first attempt to efficiently leverage pretrained text-to-image generative model to text-to-video generation model without hurting its image generation capacity. With the proposed multi-frame-rate hierarchical training framework, CogVideo is endowed with better understanding of text-video relation and ability to control the intensity of changes during generation. We extend swin attention to CogLM, which achieves acceleration in both training and inference. There are still some limitations in CogVideo, e.g. restriction on length of the input sequence still exists due to the large scale of model and limitation of GPU memory, and we leave them for future work.

**Broader Impact.** This paper aims to advance the open-domain text-to-video generation, which will ease the effort of short video and digital art creation. The efficient training method transfers knowledge from text-to-image models to text-to-video models, which helps avoid training from scratch, and thus reduce the energy consumption and carbon emission. A negative impact is the risk of misinformation. To alleviate it, we can train an additional classifier to discriminate the fakes. We believe the benefits outweigh the downsides.

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
