# OpenReview forum: "CogVideo: Large-scale Pretraining for Text-to-Video Generation via Transformers"
_NeurIPS.cc/2022/Conference — NeurIPS 2022 Submitted_

### Official Review · Reviewer_bKXQ · 2022-07-09

**Rating:** 6
**Confidence:** 4
**Ethics Flag:** Yes
**Soundness:** 3 good
**Presentation:** 3 good
**Contribution:** 3 good

**Summary:**

This paper proposed a large-scale pretrained text-to-video model and multi-frame-rate hierarchical training strategy to better align text and video clips.

**Questions:**


1. Is there a gap between the generated video and the ground truth, so ground truth is not placed in human evaluation for comparison?

2. I wonder whether the mixture factor /alpha will finally converge to 0 or 1, making the dual-channel attention block useless.


**Ethics Review Area:**

["Discrimination / Bias / Fairness Concerns", "Privacy and Security (e.g., consent)"]

**Strengths And Weaknesses:**

The strengths of this paper are as follows:

1. This paper is well written and easy to follow.
2. The proposed model is the largest and the first open-source pre-trained transformer for text-to-video generation in the general domain.
3. The experimental results show that the proposed methods can outperform a number of existing baselines.

The weaknesses of this paper are as follows:

1. The main limitation of the work is the huge consumption of GPU memory, but the related information (type and number of GPU) is not provided. I hope this information can be given.


I am not an expert in this field and do not tap into DALL-E and CogView, I will adjust the final score with other reviewers' comments and the response from the author.

---

> ### Author Response · Authors · 2022-08-02
> **Authors' response**
>
> Thanks so much for your valuable comments! CogVideo is the largest and the first open-source pre-trained transformer for text-to-video generation in the general domain, as you have mentioned in your comments. Besides, we propose dual-channel attention and multi-frame-rate training to efficiently fine-tune a pretrained text-image model to a text-video model and better align videos with text. **To further illustrate the text-to-video generation ability of CogVideo, we build a website https://cogvideo.pka.moe/**
>
> We will explain your concerns point by point:
>
> **Weakness 1:** “related information of GPU consumption.”
>
> We train the model with 104 A100 GPUs for 169,500 iterations, with batch size of 416. **The training process takes ~20 days**, and takes ~35GB memory on each GPU. In comparison, DALL-E is trained on 1024 V100 GPUs with batch size of 1024 for 430,000 iterations, and GPT-3 takes ~34 days to train if using 1,024 A100 GPUs, which is far more than the GPU consumption of CogVideo. Huge GPU consumption is indeed a common concern for large-scale pretraining, especially for text-to-video generation in general domain. That’s why we propose several optimizations in our paper, which include:
>
> 1. Dual-channel attention to efficiently inherit knowledge from pretrained text-to-image models. (Reduce the number of iterations required for training)
>
> 2. Only optimize the temporal attention channel. (Smaller memory consumption & lower computation)
>
> 3. Devise different restricted receptive fields in each attention channel, and apply swin attention. (Smaller memory consumption & accelerate computation)
>
> It is worth noting that, **without the acceleration mentioned above, it will take ~32 days (60% more)** to finish the equivalent number of optimization steps, and will converge even slower due to not inheriting the text-image knowledge.
>
>
>
> **Question 1:** “Is there a gap between the generated video and the ground truth, so ground truth is not placed in human evaluation for comparison”
>
> Yes.
>
>
>
> **Question 2:** “whether the mixture factor /alpha will finally converge to 0 or 1”
>
> We have investigated this problem in Appendix A, and visualize the mixture factor (alpha) in Figure 9. The results show that the mixture factor would converge to around 0.5~0.6, indicating both temporal and spatial channels are heavily relied on.

---

> ### Author Response · Authors · 2022-08-09
> **Kindly Reminder**
>
> Dear Reviewer: Thanks again for your careful and valuable comments! Since the rebuttal discussion is due soon, we’ll be appreciated to know whether our replies have addressed your questions. If there are any further clarifications required or any other concerns, please feel free to contact us. Many thanks!

---

### Official Review · Reviewer_xPD7 · 2022-07-11

**Rating:** 3
**Confidence:** 4
**Soundness:** 2 fair
**Presentation:** 2 fair
**Contribution:** 2 fair

**Summary:**

In this paper, the authors propose a large-scale pre-training model for text-to-video generation, called CogVideo, which is trained by inheriting a pre-trained text-to-image model CogView2. A multi-frame-rate hierarchical training strategy is designed to better align text and video clips. The proposed method is validated on standard benchmarks such as UCF-101 and Kinetics-600.

**Questions:**

1.The authors should explain the novelty and contributions of the proposed CogVideo.
2.The performance of CogVideo is not as competitive as claimed. The authors should give more discussions and analyses.
3.More details about the dataset with 5.4 million captioned videos are expected.

**Ethics Review Area:**

["I don’t know"]

**Limitations:**

Yes, the authors adequately addressed the limitations and potential negative societal impact of the work.

**Strengths And Weaknesses:**

Strengths:
1.This work explores an important topic - text-conditional video generation, and presents an effective method for it.
2.A multi-frame-rate hierarchical training strategy is proposed to improve the generation quality and control the intensity of changes during generation.
3.The authors verified the proposed model through both quantitative experiments and qualitative analyses.


Weaknesses:
1.The novelty and technical contributions of this work are quite limited. It simply assembles several existing algorithms, such as CogView2 [6] and Swin Transformer [14].
2.The performance of the proposed CogVideo is not as strong as the authors claimed. As shown in Tab. 1, the metrics of CogVideo (IS and FVD) fall behind previous methods. It is not acceptable to only compare with publicly available models.
3.The authors trained the CogVideo on a large dataset of 5.4 million captioned videos, but did not give any detailed information regarding this dataset. What are the sources of the videos and captions? How are they collected? Will the dataset be released?
4.The authors claimed that they greatly enhance parallelism and accelerate inference, but no results about inference speed of the CogVideo model are provided.
5.In Sec. 5.1, the authors "fine-tune CogVideo on the whole dataset (UCF-101) for 10,000 iterations". Is it appropriate in this field?

---

> ### Author Response · Authors · 2022-08-02
> **Authors' response (others)**
>
> Thank you very much for your valuable comments, we will explain your concerns point by point.
>
> **Weakness 2 & Question 2:** “the metrics of CogVideo (IS and FVD).”
>
> CogVideo performs well on human evaluation though, at the same time we admit that it is inferior to a few previous works on machine evaluation. This could be explained from three sides.
>
> **First**, CogVideo works for the general domain, while most existing models such as DIGAN, VideoGPT,  TATS etc. can only work for a specific domain. However, **machine evaluation is conducted on the single-domain datasets following previous work, which is unable to show the superiority of CogVideo in general domain generation**.
>
> **Second, FVD/FID and IS are actually not perfect metrics to evaluate video/image generation**. FID/FVD/IS have some suboptimality themselves. Recent studies have found that FID (similar to FVD) can be unstable to changes that humans are unaware of \[1\]. For example, a large variation of FID can be induced by differences in “low-level” processing such as image resizing and compression, which can even be of the same magnitude as the gap between different methods. It's also found that IS is sensitive to small changes that bear no relation to the quality of the generated images [4].
>
> **Third**, \[2\]\[3\] shows the inconsistency between human and machine evaluation, indicating the unreliability of FID/FVD in image/video generation tasks. Thus, **it might be more reliable to evaluate video generative models by human evaluation. To further illustrate the text-to-video generation ability of CogVideo, we have built a website https://cogvideo.pka.moe/**
>
> As many works did not release code, we are unable to perfectly reproduce their results, thus only choosing open-sourced work in human evaluation. This also exhibits the importance of open-sourcing, which is a minor contribution of CogVideo.
>
>
>
> **Weakness 3 & Question 3:** “details about the dataset”
>
> We have collected about 5.4 million text-image pairs from websites, and built a 10TB dataset (after tokenization, the size becomes about 750GB). The data is mainly crawled from professional video websites (both English and Chinese). The videos in the websites are usually with captions. About 30% of the captions are in English, which have been translated into Chinese by machine translation. About 50% of the captions constitute multiple phrases, while the other 50% are sentences. In addition, we did not remove the watermarks that appeared in some datasets even though they may affect the quality of generated images, because we think it will not influence the conclusions of our paper from the perspective of research. As the copyright of crawled images does not belong to us, currently we can't make the dataset public, but thanks for your suggestion, we will add more details about the dataset.
>
>
>
> **Weakness 4:** “experimental results about acceleration”
>
> To verify the method proposed in sec 3.3, we generate 6 frames (32*32 tokens each frame) with varying window size, and measure the time cost w/wo parallel generation. (batch size = 8)
>
> Results: https://i.imgur.com/sIr6L7E.png
>
> x-axis: shifted-window size.
>
> blue line: time cost when applying parallel generation.
>
> green line: time cost without parallel generation.
>
> As shown in the figure: (1) Applying swin attention to auto-regressive generation accelerates inference. (2) Using parallel generation can further speed up inference without affecting generated videos, and achieves around 2x acceleration when window size <= 8.
>
>
>
> **Weakness 5:** Is “fine-tuning CogVideo on the whole dataset (UCF-101) for 10,000 iterations" appropriate in this field?
>
> Yes. **Actually, it’s a common practice in the field of text-to-video generation. All models listed in Table 1 (Left) are trained on the whole UCF-101 dataset** (except those labeled by \*). Also, they usually train more steps$\times$batch-size than us.
>
> If our answers above solve your concerns, could you increase your rating?
>
>
>
> [1] Parmar, Gaurav, Richard Zhang, and Jun-Yan Zhu. "On aliased resizing and surprising subtleties in gan evaluation." Proceedings of the IEEE/CVF Conference on Computer Vision and Pattern Recognition. 2022.
>
> [2] Ramesh, Aditya, et al. "Hierarchical text-conditional image generation with clip latents." *arXiv preprint arXiv:2204.06125* (2022).
>
> [3] Ding, Ming, et al. "CogView2: Faster and Better Text-to-Image Generation via Hierarchical Transformers." arXiv preprint arXiv:2204.14217 (2022).
>
> [4] Barratt, Shane, and Rishi Sharma. "A note on the inception score." *arXiv preprint arXiv:1801.01973* (2018).

---

> ### Author Response · Authors · 2022-08-02
> **Authors' response (novelty and technical contributions)**
>
> **Regarding the novelty and technical contributions of this work.**
>
> Res: This might be a misunderstanding. The techniques behind the proposed CogVideo is not that straightforward. We have carefully tested each of them and found, without each of them, the video generation performance will be significantly reduced. To summarize, as stated in the Introduction section, there are three main contributions. Here, we will stress why the three contributions are important.
>
> - **Contribution 1:** Multi-frame-rate hierarchical training framework.
>
>   Most existing works model videos at a fixed frame rate and length, which makes it very easy to result in misalignment between texts and videos because of the varying duration of videos (see L32-36 for an example). By modeling the video together with the frame rate, CogVideo successfully ensures the alignment of text-video pairs, thus better at capturing the relationship between texts and videos, and improves FVD by 10% according to the ablation study. In addition, the multi-frame-rate hierarchical training method also offers CogVideo the capacity of controlling the intensity of changes during generation.
>
> - **Contribution 2:** Using Dual-channel Attention, CogVideo is the first text-to-video generative model finetuned from a pretrained text-to-image transformer (to the best of our knowledge).
>
>   During experiments, we found that pretraining a large text-to-video model in the general domain usually suffers from severe lack of data. That’s part of the reason why existing SOTA models mostly work for a specific domain. In the meantime, many successful text-to-image pretrained transformers that can be beneficial to text-to-video pretraining remain unexplored.
>
>   We propose dual-channel attention as a universal approach to efficiently inherit knowledge from pretrained models without catastrophic forgetting. With effects of 1) Alleviating data scarcity and 2) reducing computational resources, it improves FVD by 30% according to the ablation study.
>
>   **It can also be applied in other tasks and modalities,** such as finetuning an image generative model to a super-resolution model, finetuning a text-to-image generative model to an audio-to-image generative model, etc.
>
> - **Contribution 3:** CogVideo performs text-to-video generation in general domain.
>
>   As the largest pretrained transformer for text-to-video generation, CogVideo is able to work in different domains including but not limited to people, animals, objects, landscapes, etc. In comparison, most existing SOTA models such as DIGAN, VideoGPT, and TATS can only work for a specific domain. We have open-sourced the code and pretrained model to benefit the development of this topic.
>
> Other contribution includes adopting Shifted-window attention to the auto-regressive scenario, which accelerates both training and inference.

---

> ### Author Response · Authors · 2022-08-09
> **Kindly Reminder**
>
> Dear Reviewer: Thanks again for your careful and valuable comments! Since the rebuttal discussion is due soon, we’ll be appreciated to know whether our replies have addressed your questions. If there are any further clarifications required or any other concerns, please feel free to contact us. If our replies above solve your concerns, could you please increase your rating? Many thanks!

---

### Official Review · Reviewer_WUEH · 2022-07-11

**Rating:** 6
**Confidence:** 4
**Soundness:** 3 good
**Presentation:** 2 fair
**Contribution:** 3 good

**Summary:**

This paper is the first work to propose an open-source pretrained transformer to solve the text-to-video task. The authors propose a two-stage framework CogVideo by finetuning a text-to-image model and avoid pretraining from scratch to reduce the training cost. The idea of multi-frame-rate ensures the flexibility and accuracy of the generated video. Experiments and visualized samples show the effectiveness of the method on video generating.

**Questions:**

1)Is the fps a learnable parameter or a hyper-parameter? And how to use the fps token concretely in the transformer?

2)Compared with other training method, does CogVideo accelerate the training process? What environment and how much time does the training process need?

3)Compared with the 1-Stage finetuned CogVideo model, how does the full model deal with the problem that one generated frame in the sequential generation stage collapses and influences the later frames? I do not have any idea that the two stage pipeline can solve this problem.

**Limitations:**

No.
And I do not see any serious concerns.

**Strengths And Weaknesses:**

（1）Strengths：

The paper proposes the pretrained two-stage Sequential Generation and Recursive Interpolation pipeline and better reconstruct the alignment relation in a video. And the two-stage method does help the model converge better and faster, which is proved by the line chart of training loss shown in the ablation studies. I think the CogVideo pipeline has its value in real-world application scenarios. Also, the paper is well writing.

（2）Weaknesses:

-First, the paper is not the first attempting to investigate hierarchical transformer structure in text-to-video challenge. For example, [1] also exploits autoregressive and interpolation transformers and gain better performance on UCF-101. However, the authors do not explain the difference or innovation compared with [1].
-The authors have claimed that the CogVideo pipeline can solve the text-to-video generation in the general domain, but the experiment results in tab.1 are all attained on human action video datasets( UCF-101 and Kinetics-600), which maybe somewhat weak to prove the above conclusion. I guess maybe general domain represents the motions in the text have some temporal relation. If the concept of general domain can be defined in the paper, it will be much better.

[1] S. Ge, T. Hayes, H. Yang, X. Yin, G. Pang, D. Jacobs, J.-B. Huang, and D. Parikh. Long video generation with time-agnostic vqgan and time-sensitive transformer. arXiv preprint arXiv:2204.03638, 2022

---

> ### Author Response · Authors · 2022-08-02
> **Authors' response**
>
> Thank you very much for your valuable comments, we will explain your concerns point by point.
>
> **Weakness 1:** “the difference or innovation compared with [1]”
>
> The **motivation** (Line 32-36, 40-45) and **method** (section 3.1) of the multi-frame-rate hierarchical framework in CogVideo are very different from that in [1]. While [1] focuses on generating longer videos, our motivation is to ensure the alignment between text and its temporal counterparts in the video, which is the common concern in modeling text-video relationships as the duration of videos varies a lot. As for the method, besides being hierarchical, we additionally prepend a piece of text describing the frame rate to the original text prompt to joint model video and frame rate, and introduce CogLM to better explore the bidirectional context.
>
> Besides, **CogVideo works for the general domain, while [1] can only work for a specific domain**. However, machine evaluation is conducted on the single-domain datasets following previous work, which is unable to show the superiority of CogVideo in general domain generation.
>
>
> **Weakness 2:**
>
> - “the experiment results in tab.1 are all attained on human action video datasets (UCF-101 and Kinetics-600), which maybe somewhat weak to prove the above conclusion”
>
>   We follow the experiment settings of previous work in order to compare with them. As the reviewer has pointed out, **the datasets fail to show the superiority of CogVideo in general domain generation, which is actually a drawback of the existing evaluation protocol**, not a weakness of CogVideo.
>
> - "the ability to perform text-to-video in general domain"
>
>   CogVideo can generate samples covering multiple domains including but not limited to people, animals, objects, landscapes, etc.
>
>   **More samples**: https://i.imgur.com/OPUc6O4.jpg (besides samples in Figure 1 (main text), Figure 11 (appendix) ).
>
>   To further illustrate the text-to-video generation ability of CogVideo, **we build a website** https://cogvideo.pka.moe/.
>
>   The domain that the pretrained model can perform generation in is actually data defined. As our dataset is collected from the general domain (covering people, animals, objects, landscapes, weather, comics, etc. ), CogVideo is able to perform text-to-video generation in general domain.
>
>
>
> **Question 1:**
>
> Fps is a hyper-parameter. For Stage 1, it is determined by the length of each training sample so that it accommodates the whole sample and ensures text-video alignment. For Stage 2, we manually choose several frame rates (2, 4, 8 fps).
>
> To be concrete, we prepend a piece of text describing the frame rate to the original text prompt, thanks to the flexibility of the textual condition.
>
>
>
> **Question 2:**
>
> - “ Does CogVideo accelerate the training process”
>
>   Yes. we propose several ways to accelerate, which include:
>
> 1. Dual-channel attention to efficiently inherit knowledge from pretrained text-to-image models. (Reduce the number of iterations required for training)
>
> 2. Only optimize the temporal attention channel. (Smaller memory consumption & lower computation)
>
> 3. Devise different restricted receptive fields in each attention channel, and apply swin attention to the autoregressive scenario. (Smaller memory consumption & accelerate computation)
>
> - “ What environment and how much time does the training process need?”
>
>   We train the model with batch size of 416 on 104 A100 GPUs. The training process takes ~20 days. Without the acceleration mentioned above, it will take ~32 days (60% more) to finish the equivalent number of optimization steps, and will converge even slower due to not inheriting the text-image knowledge.
>
>
>
> **Question 3:** “how does CogVideo deal with the problem that one generated frame in the sequential generation stage collapses and influences the later frames”
>
> More cases comparing hierarchical and 1-Stage generation: https://i.imgur.com/vC8SowL.png
>
> A video contains a large number of frames. Sequentially generating all frames is proven to work well in earlier frames, but often causes severe quality degradation in later frames, e.g. Fig 4,12 in [2]. To deal with it, our framework only generates a small number (we choose 5) of frames in Stage 1, thus avoiding severe quality degradation and globally ensuring the video quality. Then in Stage 2, given the high-quality frames generated in Stage 1, the model is able to interpolate frames with minor quality degradation.
>
>
>
> [1] Ge, Songwei, et al. "Long video generation with time-agnostic vqgan and time-sensitive transformer." arXiv preprint arXiv:2204.03638 (2022).
>
> [2] Rakhimov, Ruslan, et al. "Latent video transformer." arXiv preprint arXiv:2006.10704 (2020).

---

> ### Author Response · Authors · 2022-08-09
> **Kindly Reminder**
>
> Dear Reviewer: Thanks again for your careful and valuable comments! Since the rebuttal discussion is due soon, we’ll be appreciated to know whether our replies have addressed your questions. If there are any further clarifications required or any other concerns, please feel free to contact us. Many thanks!

---

### Official Review · Reviewer_fg6U · 2022-07-16

**Rating:** 6
**Confidence:** 4
**Ethics Flag:** Yes
**Soundness:** 2 fair
**Presentation:** 3 good
**Contribution:** 2 fair

**Summary:**

This work addresses the large-scale pre-training in the open-domain Text-to-Video Generation. The major challenge to this task is the lack of large text-video paired datasets available and the weak relevance between them. In this paper, the authors propose a 9B-parameter transformer model CogVideo which is trained by inheriting the learned spatial semantics from a pre-trained text-to-image model (CogView2 in this paper).

They claim that the key difference between the text-to-video generation and the text-to-image generation is that the former needs huge of paired data to infer both spatial and temporal correlation between two modalities while the latter only requires the learning of spatial correlation. Therefore, they propose a dual-attention channel that adds an additional attention layer based on original structure of CogView2 to address the learning of temporal correlation (as illustrated in Figure 3). During training, it only optimizes the parameters of newly added temporal attention layer while keep all the parameters of CogView2 frozen.

To address the weak alignment of text and variable-length video (the example in Line 32-36 gives a good illustration of this issue), they propose multi-frame-rate hierarchical training. Concretely, it involves a two-stage generation process. In stage 1, they propose to add a frame-rate token to the text to generate the image frames at low frame rate. Then in stage 2, they introduce another frame interpolation transformer to generate the immediate transition frames between the generated frames of the first transformer model in stage 1.

The proposed hierarchical generation framework looks promising on long videos given a text input. The idea of interpolation transformer model that can utilize bidirectional frame context to finish the interpolation of current frame also make sense. However, there are some critical weaknesses and questions listed below.


**Questions:**

**Questions for the authors:**
1) Why are there no detailed analysis about main experimental results in Table 1 in the main text of the paper?
2) It seems there are only an ablation study on Kinetics-600. Why not including the ablation study on UCF-101 datasets in Section 5.3? It would be more convincing to cover the comparison results on UCF-101.
3) In Figure 7, it looks there is no obvious performance gap between (a) and (e). Thus, what’s the advantage of hierarchical generation from the perspective of visualization results, not just in human evaluation results. Could you provide more visualized case analysis in appendix later to justify the superior performance of your proposed method?
4) In Section 3.2, Is the restricted receptive field formulation in Eq. 4 like the 3D nearby attention mechanism in NUWA [36]? If yes, what’s the difference between two?
5) How many seconds do the longest videos that your method generate can last? In my understanding, this is the key advantage of the hierarchical generation framework.
6) How to understand the recursive process of hierarchical generation? Do you mean that immediate interpolated (generated) frames at each iteration will be used to generate new immediate frames between previous ones? Why not directly generating all immediate frames at once?

**Minor issues:**
1) Could you please add a citation or hyperlink for DeepSpeed in the paper later? Since it is a great work in the community, you should mention it at least.
2) Grammar error in Line 287, “samples generated hierarchically performs…” => “samples … perform …”.



**Ethics Review Area:**

["Discrimination / Bias / Fairness Concerns"]

**Limitations:**

The major limitation is that how long the generated videos by the proposed CogVideo can last. The submitted demos show that the generated videos can only last several seconds. It would be more exciting that CogVideo can generate longer videos for given text input.

**Strengths And Weaknesses:**

**Strengths:**
1) The proposed multi-frame-rate hierarchical training has the potential on the text-to-video generation, especially the idea of frame interpolation transformer model that can utilize the bidirectional frame context to generate immediate frame. It indeed looks a promising solution to handle long video generation given a text input.
2) The proposed CogVideo inherits the parameters of a pre-trained text-to-image models and avoid the expensive pre-training from scratch.
3) In addition, the authors claim that they will open-source the proposed large-scale text-to-video generation model. This is another contribution to the community.

**Weaknesses:**
1) In the main experiment results in Table 1, it seems that CogVideo obviously underperforms TATS-base [9] on UCF-101 dataset and TriVD-GAN-FP [16] on Kinetics-600. This conflicts with the claim in abstract. Is that due to the issue of metric itself or other reasons? Could you explain more about results in Table 1 since I cannot find related information in the main text?
2) It is not clear how much the temporal attention channel contributes to the performance gains of CogVideo. In the ablation study, it does not include the comparison of with/without temporal attention layer, i.e., compare the performance gap of directly fine-tuning CogView2’s weights and keeping CogView2’s weights frozen while tuning the temporal attention layer.
3) The qualitative evaluation only includes one case analysis, it would be more convincing to provide more visualized comparison between different variants. Since the visualization samples are much more intuitive than evaluation metrics to directly judge the real performance of the model.

---

> ### Author Response · Authors · 2022-08-02
> **Authors' response (for weaknesses)**
>
> Thank you very much for your careful and valuable comments. We will explain your concerns point by point. Some results you wanted (e.g. Weakness 2, 3) were originally in the paper and were removed from our draft or moved to the appendix due to limitations of space. Here we show the results and will add them back to the camera-ready version if accepted.
>
> **Weakness 1:** “explain more about the results in Table 1”
>
> CogVideo performs well on human evaluation though, at the same time we admit that it is inferior to a few previous works on machine evaluation. This could be explained from three sides.
>
> **First**, CogVideo works for the general domain, while most existing models such as DIGAN, VideoGPT, TATS etc. can only work for a specific domain. However, **machine evaluation is conducted on the single-domain datasets following previous work, which is unable to show the superiority of CogVideo in general domain generation**.
>
> **Second, FVD/FID and IS are actually not perfect metrics to evaluate video/image generation**. FID/FVD/IS have some suboptimality themselves. Recent studies have found that FID (similar to FVD) can be unstable to changes that humans are unaware of \[1\]. For example, a large variation of FID can be induced by differences in “low-level” processing such as image resizing and compression, which can even be of the same magnitude as the gap between different methods. It's also found that IS is sensitive to small changes that bear no relation to the quality of the generated images [4].
>
> **Third,** \[2\]\[3\] shows the inconsistency between human and machine evaluation, indicating the unreliability of FID/FVD in image/video generation tasks. Thus, **it might be more reliable to evaluate video generative models by human evaluation. To further illustrate the text-to-video generation ability of CogVideo, we have built a website https://cogvideo.pka.moe/**
>
> As many works did not release code, we are unable to perfectly reproduce their results, thus only choosing open-sourced work in human evaluation. This also exhibits the importance of open-sourcing, which is a minor contribution of CogVideo.
>
>
>
> **Weakness 2**: “how much the temporal attention channel contributes to the performance gains of CogVideo”
>
> 1. We’ve investigated the dual-channel attention in Appendix. A, which shows that both spatial and temporal channels are relied on.
>
> 2. Following your suggestion, we additionally compare fine-tuning with/without the temporal attention channel.
>
> | Method                                        | Setting                                                      | FVD    |
> | --------------------------------------------- | ------------------------------------------------------------ | ------ |
> | CogVideo                                      | None                                                         | 108.27 |
> | Finetuned CogView2 (with temporal channel)    | keeping CogView2’s weights frozen while tuning the temporal attention layer. | 124.92 |
> | Finetuned CogView2 (without temporal channel) | directly fine-tuning CogView2’s weights                      | 276.57 |
>
> As shown in the table, directly fine-tuning CogView2 is worse than keeping CogView2’s weights frozen while tuning the temporal attention layer. During the experiment of directly fine-tuning CogView2’s weights, we find that loss tends to be high for a long time in the initial phase of training, which implies that parameters are greatly changed and a lot of knowledge stored in CogView2 is lost. On the contrary, our method avoids this problem by freezing the spatial channel to CogView2.
>
> For qualitative case, please refer to the answer to Weakness 3.
>
>
>
> **Weakness 3:**
>
> More qualitative cases: https://i.imgur.com/RemXm1Y.jpg
>
> As shown in the image, videos hierarchically generated by finetuned CogVideo (1) have frames of higher quality (e.g. the human's shape), especially in the later frames; (2) the movement is more reasonable and consistent with the text.
>
>
> [1] Parmar, Gaurav, Richard Zhang, and Jun-Yan Zhu. "On aliased resizing and surprising subtleties in gan evaluation." Proceedings of the IEEE/CVF Conference on Computer Vision and Pattern Recognition. 2022.
>
> [2] Ramesh, Aditya, et al. "Hierarchical text-conditional image generation with clip latents." *arXiv preprint arXiv:2204.06125* (2022).
>
> [3] Ding, Ming, et al. "CogView2: Faster and Better Text-to-Image Generation via Hierarchical Transformers." arXiv preprint arXiv:2204.14217 (2022).
>
> [4] Barratt, Shane, and Rishi Sharma. "A note on the inception score." *arXiv preprint arXiv:1801.01973* (2018).

---

> ### Author Response · Authors · 2022-08-02
> **Authors' response (for questions)**
>
> **Question 1:** “detailed analysis about main experimental results in Table 1”
>
> Please refer to the answer to weakness 1.
>
>
>
> **Question 2:**
>
> We follow VideoGPT\[5\], Video Transformer\[6\], Latent Video Transformer\[7\] to conduct the ablation study on one dataset. As Kinetics-600 is larger than UCF-101 and thus more convincing, we choose Kinetics-600.
>
>
>
> **Question 3:**
>
> More cases of the comparison between hierarchical and sequential generation (without finetuning on Kinetics-600) : https://i.imgur.com/vC8SowL.png
>
> As shown in the image, hierarchically generated videos tend to be more stable (e.g. better shape of the objects) and aligned with the texts (e.g. the process of drinking coffee and singing).
>
>
>
> **Question 4:**
>
> There are two main differences between us and NUWA[8]:
>
> 1. We adopt varying sizes of restricted receptive fields in different attention channels. While maintaining the ability to cut down computation, our formulation of receptive fields allows adequate exploration of temporal/spatial relationships in the temporal/spatial channel.
>
> 2. We write custom CUDA kernels to accelerate the local attention.
>
>
>
> **Question 5:**
>
> - “How many seconds do the longest videos that your method generates can last”
>
>   Theoretically, the multi-frame-rate hierarchical framework allows video generation of any length. In practice, we observed that most actions last within 4 seconds, and longer videos might introduce more noise (repetition of actions / fuzzy text-video relationship). Therefore, we set the maximum length to 4 seconds in our pretraining.
>
> - “The key advantage of the hierarchical generation framework”
>
>   The key advantage of our multi-frame-rate hierarchical framework is to keep text-video alignment.
>
>
>
> **Question 6:**
>
> - “How to understand the recursive process of hierarchical generation? Do you mean that immediate interpolated (generated) frames at each iteration will be used to generate new immediate frames between previous ones"
>
>   Yes
>
> - “Why not directly generate all immediate frames at once”
>
>   Recursive generation provides a simple yet flexible way to interpolate videos of different frame rates. For example, if a 1/2 fps video has been generated on stage1, and we want to interpolate it into 4/8 fps. With the multi-frame-rate framework, it can be achieved by simply adjusting the frame-rate token and the number of times performing Stage 2. However, if generating all immediate frames at once, the model has to learn multiple tasks of interpolating 1/3/7 frames between every two frames at different frame rates. Also, the maximum frame rate is restricted by GPU memory.
>
>
>
> If our answers above solve your concerns, could you increase your rating?
>
> [5] Yan, Wilson, et al. "Videogpt: Video generation using vq-vae and transformers." arXiv preprint arXiv:2104.10157 (2021).
>
> [6] Neimark, Daniel, et al. "Video transformer network." Proceedings of the IEEE/CVF International Conference on Computer Vision. 2021.
>
> [7] Weissenborn, Dirk, Oscar Täckström, and Jakob Uszkoreit. "Scaling autoregressive video models." arXiv preprint arXiv:1906.02634 (2019).
>
> [8] Wu, Chenfei, et al. "N\" uwa: Visual synthesis pre-training for neural visual world creation." arXiv preprint arXiv:2111.12417 (2021).

---

> ### Author Response · Authors · 2022-08-09
> **Kindly Reminder**
>
> Dear Reviewer: Thanks again for your careful and valuable comments! Since the rebuttal discussion is due soon, we’ll be appreciated to know whether our replies have addressed your questions. If there are any further clarifications required or any other concerns, please feel free to contact us. Many thanks!

---

> > ### Comment · Reviewer_fg6U · 2022-08-09
> > **Thank you for your detailed responses.**
> >
> > Thank the authors for addressing my concerns. After reading your response and other peer reviews, I decide to raise my score to 6.

---

### Review · Ethics_Reviewer_7NSB · 2022-08-04

**Recommendation:**

Adding information from the authors' comment to the paper itself would be an improvement.

Adding information about the ethics of the data sets and their use would be an improvement.

If this additional information checks out, I think the submission will be ethically acceptable.


**Ethical Issues:**

Yes

**Ethics Review:**

Two reviews flagged this submission for an ethics review.  Both flagged it as raising issues about "Discrimination / Bias / Fairness Concerns".  One also flagged it as  raising issues about "Privacy and Security (e.g., consent)".  Both reviews do not say exactly what ethics concerns were raised by the submission.

The authors acknowledged the fairness concern.  So, I deal with it below.

I'm guessing the concern about "Privacy and Security (e.g., consent)" is about the data sets used.  They contain videos of people.  Was the training data collected in an ethical manner?  Did these people give consent for the videos to be used in this way?  If not, is it OK anyhow?  Is using videos in this way violating their privacy?  For example, do the people in videos generated by the submission's system look like real people in the training data?

The issues should be addressed as discussed under the headings

 - "1. Contain any personally identifiable information or sensitive personally identifiable information."

 - "1. Consent to use or share the data."

 - "7. Deceive people in ways that cause harm."

of the Ethics Guidelines at https://neurips.cc/public/EthicsGuidelines

I checked the papers cited for one of the used data sets, the one cited as [23].  I didn't notice any discussion of ethical issues in it.  So, the authors of this submission might need to get in touch with the authors of that paper to answer these questions.

---

### Review · Ethics_Reviewer_6SsL · 2022-08-06

**Recommendation:**

I think the only way to address the concerns in the current version of the paper is to clearly state and acknowledge them.
The proposed "solution" is not a solution, and should not be claimed as such.

**Ethical Issues:**

Yes

**Ethics Review:**

The work presents a large text-to-video model.

It inherits some of the ethical concerns of the text-to-image generation models, where, for example, the generated image may be perpetuating or amplifying existing stereotypes along racial or gender lines.

---

### Meta-Review · Program_Chairs · 2022-08-26

**Recommendation:** Reject
**Confidence:** Less certain

**Metareview:**

The authors address an important problem of text-to-video generation in this work. Although there is a set of solid contributions that are specific to video generation, author-reviewer discussion as well as the following reviewer-area chair discussion did not address all the concerns raised by the reviewers. Furthermore, the lack of and also superficial address of the ethical concerns raised by the reviewers as well as ethics reviewers strongly suggests that the authors need to substantially revise the manuscript along both technical and ethical aspects. The paper is thus recommended to be rejected.

**Award:**

No

---

### Decision · Program_Chairs · 2022-09-14

Reject